# Risk of Nipah Virus Seroprevalence in Healthcare Workers: A Systematic Review with Meta-Analysis

**DOI:** 10.3390/v17010081

**Published:** 2025-01-09

**Authors:** Matteo Riccò, Antonio Cascio, Claudio Costantino, Silvia Corrado, Ilaria Zanella, Pasquale Gianluca Giuri, Susanna Esposito

**Affiliations:** 1AUSL–IRCCS di Reggio Emilia, Servizio di Prevenzione e Sicurezza Negli Ambienti di Lavoro (SPSAL), Local Health Unit of Reggio Emilia, 42122 Reggio Emilia, Italy; 2Infectious and Tropical Diseases Unit, Department of Health Promotion, Mother and Child Care, Internal Medicine and Medical Specialties, “G. D’Alessandro”, University of Palermo, AOUP P. Giaccone, 90127 Palermo, Italy; antonio.cascio03@unipa.it; 3Department of Health Promotion Sciences, Maternal and Infant Care, Internal Medicine and Medical Specialties (PROMISE) “G. D’Alessandro”, University of Palermo, 90127 Palermo, Italy; claudio.costantino01@unipa.it; 4ASST Rhodense, Dipartimento della Donna e Area Materno-Infantile, UOC Pediatria, 20024 Milan, Italy; scorrado@asst-rhodense.it; 5Department of Medicine and Diagnostics, AUSL di Parma, 43100 Parma, Italy; pgiuri@ausl.pr.it; 6Pediatric Clinic, Department of Medicine and Surgery, University of Parma, 43126 Parma, Italy; susannamariaroberta.esposito@unipr.it

**Keywords:** Nipah virus (NiV), bats, control, diagnosis, encephalitis, epidemiology, pathology, prevention, therapeutics, vaccines, zoonosis

## Abstract

Nipah virus (NiV) is a zoonotic pathogen with the potential to cause human outbreaks with a high case fatality ratio. In this systematic review and meta-analysis, available evidence on NiV infections occurring in healthcare workers (HCWs) was collected and critically appraised. According to the PRISMA statement, four medical databases (PubMed, CINAHL, EMBASE, and Scopus) and the preprint repository medRixv were inquired through a specifically designed searching strategy. A total of 2593 entries were identified; of them, 16 studies were included in qualitative and quantitative analysis detailing the outcome of NiV infection on HCWs and estimates of seroprevalence among healthcare professions. All studies reported data from Asian countries: Malaysia, Singapore, Bangladesh, India (States of Kerala and Bengal), and Philippines. Seroprevalence was estimated from seven studies in 0.00% (95%CI 0.00 to 0.10) for IgM-class antibodies and 0.08% (95%CI 0.00 to 0.72) for IgG class-antibodies, but four of the sampled studies did not report any seropositive cases. A case fatality ratio of 73.52% (95%CI 34.01 to 99.74) was calculated from 10 studies. In conclusion, the present study shows that NiV may result in a possible occupational infection among HCWs involved in managing incident cases. As most NiV outbreaks occur in limited resources settings, it is reasonable that even basic preventive measures (i.e., mandatory use of PPE and appropriate isolation of incident cases with physical distancing) may be quite effective in avoiding the occurrence of new infections among HCWs.

## 1. Introduction

Nipah virus (*Henipavirus nipahense*, NiV) is an enveloped RNA virus belonging to the genus *Henipavirus* (according to the current classification of the International Committee on Taxonomy of Viruses [ICTV]: order: *Mononegavirales*; family: *Paramixoviridae*; subfamily: *Orthoparamyxovirinae*) [1,2,3]. The viral particle of NiV is highly pleomorphic, with a shape ranging from spherical to filamentous, a diameter ranging from 40 to 1900 nm, and a nucleocapsid core, which includes a single-stranded, negative-sense RNA genome (18.2 kbases) that contains a total of six genes from 3′ to 5′: nucleocapsid (N), phosphoprotein (P), matrix (M), fusion glycoprotein (F, the processed by host protease in two subunits, F1 and F2), attachment glycoprotein (G), and the long polymerase or RNA-dependent RNA polymerase (L) [4,5]. In turn, the P gene, through post-transcriptional editing and alternative start codons, encodes three non-structural proteins, C, V, and W, which are involved in the pathogenesis of NiV infection by inhibiting interferon production and signaling in the host cells [2,5].

Human infections due to NiV have been described since late September 1998, when a cluster of patients associated with pig farming from the suburb of Ipoh city, within the Malaysian state of Perak, presented with signs and symptoms of acute febrile encephalitis (i.e., fever, headache, and an altered state of consciousness), with high mortality (15 deaths over 27 total cases, case fatality ratio [CFR] of 55.56%) [5,6,7,8,9]. The human outbreak was preceded by the occurrence of respiratory illness and encephalitis in pigs from the same district [7]. In the following months, several clusters of febrile encephalitis were reported from other areas, encompassing a total of 283 documented cases and 109 deaths [5,7,8,9]. Eventually, a new virus (NiV) was isolated from the cerebrospinal fluid (CSF) of a patient from Sungai Nipah village, receiving its current name [6,7,8], and is thought to be characterized as a zoonotic pathogen with a wide range of amplification hosts (i.e., pig, dog, cat, horse, hamster, guinea pig, bat, and ferret) [10]. The NiV reservoir was subsequently identified in fruit bats or flying foxes (bats in the *Pteropus* genus, unevenly distributed across the coastal regions and in several islands in the Indian Ocean, India, Southeast Asia, and Oceania), which disseminate the virus by natural secretions and excreta (i.e., saliva, urine, feces, etc.) [2]. In fact, *Henipavirus* is to date considered the only genus from the family of *Paramyxoviridae* including zoonotic species [11]. According to the available reconstruction of the spillover event, flying foxes were attracted by the local commercial fruit production industry, enabling the contact of livestock with chewed fruits and bat excreta, leading to the initial transmission in the naïve pig population, then prompting human infections initially in occupational settings (i.e., among people involved in farming and meat processing) [2,7,12,13]. During the first outbreaks, inter-human transmission was not documented [14]. Since 1998, the NiV has surfaced in other countries of Southeast Asia, initially from livestock sales [2], starting from Singapore in 1999 [15], then involving central and north-western Bangladesh in 2001 [5,14], Philippines in 2014 [16], and most notably India, initially from West Bengal (2001 and 2007), then from Kerala, a southern state on the west coast [5,6,14,17,18,19,20,21,22]. Notably, epidemics from Bangladesh and Kerala have been sustained by a different strain of NiV (NiV-B) from that initially identified in Malaysia and Singapore (NiV-A), characterized by the direct transmission of the pathogen from bats to fruits and from fruits to humans, with subsequent interhuman spreading by contact with their body fluids, or through aerosols [4,5,23,24,25,26], in a series of repeated outbreaks, which in Bangladesh have developed a seasonal pattern (associated with the harvesting season of date palm sap) with very high lethality (around 75%) [2,5,21,27].

After the interaction of respiratory epithelium with aerosols and respiratory droplets contaminated with NiV particles, the pathogen replicates in respiratory cells then invade the stromal cells and eventually the endothelia of the pulmonary vascular bed [2], and then disseminate to other tissues, the most notable being the central nervous system. According to our current understanding, asymptomatic and subclinical cases are rare, at less than 20% [7]. Usually, human NiV infection initially causes a flu-like clinical syndrome (day +7 to +11), complicated by respiratory symptoms such as sore throat, cough, or respiratory distress [5,7,8,15,28], then progressing to a severe, rapidly progressive encephalitis characterized by brain stem dysfunction with a high case–fatality ratio [2,4,5,17,24]. In their recent meta-analysis, Vasudevan et al. [24] identified a CFR of 80.1% (95% confidence intervals [95%CI] 68.7 to 88.1), ranging from 82.7% (95%CI 74.6 to 88.6) for India, followed by Bangladesh (62.1%, 95%CI 45.6 to 76.2%) and Philippines (52.9%, 95%CI 30 to 74.5). Moreover, around 22% of survivors develop a neurological deficit [2], including cerebellar signs, either tetraparesis or monoparesis, cranial nerve palsies, peripheral nerve lesions, and even higher mental function deficits [9]. However, it is important to stress that most available studies on NiV infections have been designed initially as observational reports from documented outbreaks; a common issue of studies on emerging diseases is the potential overestimation of the actual lethality due to potential sampling biases associated with local capabilities for epidemiological surveillance and clinical management [11,24,29].

Due to the reportedly high mortality rate and the potential interhuman spreading of NiV-B, NiV represents a serious public health concern, particularly from the point of view of healthcare workers (HCWs). As recently stressed by Cubelo et al. [28], not only are HCWs on the frontline during NiV outbreaks, facing the risks associated with a pathogen with high lethality [6,22,30,31], but there is some evidence that patients may have contracted NiV either from index cases and/or from HCWs involved in the care of index cases within healthcare facilities [26,30,31,32].

In other words, even though the potential for the global spread of NiV still remains quite limited [5,23,25,26,28], it could represent a challenging effort for HCWs and healthcare systems [28,33,34], as recently stressed [35] not only from the health safety point of view, but also when considering the potential ethical issues. Due to the perceived but still latent risk that NiV may evolve from an emerging tropical disease to a potentially epidemic or even pandemic global threat [36,37,38], assessing how this pathogen affected HCWs and healthcare systems during local outbreaks may provide some guidance for all healthcare professionals and stakeholders. Therefore, this systematic review with a meta-analysis aimed to gather and synthesize the existing evidence on the seroprevalence of NiV in HCWs involved in the management of index cases. Our research could provide a comprehensive understanding of the risks possibly associated with inpatient care during an outbreak, leading to developing strategies for improving preparedness, safety, and general support during NiV epidemics.

## 2. Materials and Methods

### 2.1. Research Concept

The present systematic review with meta-analysis was designed in accordance with the PRISMA statement (Prepared Items for Systematic Reviews and Meta-Analysis) [39,40,41] in order to ascertain the seroprevalence (IgM and/or IgG) of NiV infection among HCWs involved in the management of index cases, and the research concept was in accord with the “PICO” strategy (Patient/Population/Problem, Investigated results, Control/Comparator, and Outcome) as follows:Population of interest: HCWs involved in the management of documented cases of NiV;Investigated result(s): exposure to NiV infections in terms of seroprevalence (IgM and/or IgG) AND/OR documented infections AND/OR their outcome (where available);Control: HCWs from the same healthcare facilities not involved in the management of NiV infections.Outcome: seroprevalence AND/OR documented mortality and short/long term complications.

The present study protocol was preventively registered into the PROSPERO database (Prospective Register of Systematic Reviews) with the ID number CRD42024594832 (Appendix A).

### 2.2. Study Selection, Inclusion and Exclusion Criteria

Starting on 25 September 2024, the following databases were searched for entries on NiV without any backward chronological restriction by applying a research strategy adapted to the specificities of the inquired database (Appendix B, Table A1).

PubMed: “Nipah Virus”[Mesh] AND ((“Indonesia” OR “Cambodia” OR “Timor” OR “Malaysia” OR “Philippines” OR “Singapore” OR “Thailand” OR “India” OR “Bangladesh”) OR (“Health Personnel” [Mesh] OR “Allied Health Personnel” [Mesh] OR “healthcare worker*” OR “health care worker*” OR “nurs*” OR “work*” OR “occupational” OR “health professional*” OR “medical practitioner*” OR “medical doctor*” OR “nursing professional*” OR “midwifery professional*” OR “midwife” OR “paramedic*” OR “surgical technician*” OR “dentist*” OR “physiotherapist*” OR “laboratory technician*” OR “pathologist*” OR “medical assistant*” OR “ambulance officer*” OR “emergency medical technician*” OR “emergency paramedic*”)).

Scopus: “Nipah” OR “Nipah virus” AND ((“Indonesia” OR “Cambodia” OR “Timor” OR “Malaysia” OR “Philippines” OR “Singapore” OR “Thailand” OR “India” OR “Bangladesh”) OR (“Health Personnel” OR “Allied Health Personnel” OR “healthcare worker*” OR “health care worker*” OR “nurs*” OR “work*” OR “occupational” OR “health professional*” OR “medical practitioner*” OR “medical doctor*” OR “nursing professional*” OR “midwifery professional*” OR “midwife” OR “paramedic*” OR “surgical technician*” OR “dentist*” OR “physiotherapist*” OR “laboratory technician*” OR “pathologist*” OR “medical assistant*” OR “ambulance officer*” OR “emergency medical technician*” OR “emergency paramedic*”))EMBASE: (‘nipah virus’/exp OR ‘nipah virus’ OR ‘nipah virus infection’) AND (‘health care personnel’ OR ‘occupational’ OR ‘work AND related’ OR ‘nursing staff’ OR ‘nurse’).

CINAHL: “Nipah” AND ((“Indonesia” OR “Cambodia” OR “Timor” OR “Malaysia” OR “Philippines” OR “Singapore” OR “Thailand” OR “India” OR “Bangladesh”) OR (“Health Personnel” OR “Allied Health Personnel” OR “healthcare worker*” OR “health care worker*” OR “nurs*” OR “work*” OR “occupational” OR “health professional*” OR “medical practitioner*” OR “medical doctor*” OR “nursing professional*” OR “midwifery professional*” OR “midwife” OR “paramedic*” OR “surgical technician*” OR “dentist*” OR “physiotherapist*” OR “laboratory technician*” OR “pathologist*” OR “medical assistant*” OR “ambulance officer*” OR “emergency medical technician*” OR “emergency paramedic*”)).

EMBASE: (‘nipah virus’/exp OR ‘nipah virus’ OR ‘nipah virus infection’) AND (‘health care personnel’ OR ‘occupational’ OR ‘work AND related’ OR ‘nursing staff’ OR ‘nurse’ OR ‘occupational accident’ OR ‘paramedical personnel’ OR ‘health practitioner’ OR ‘midwife’ OR ‘physiotherapist’ OR ‘rescue personnel’ OR ‘laboratory personnel’ OR ‘medical assistant’ OR ‘dentist’ OR ‘pathologist’).

MEDRXIV: “Nipah” OR “nipah virus” AND (“health personnel” OR “allied health personnel” OR “healthcare worker*” OR “nurs*”).

In order to gather all available evidence, a “snowball” approach was also applied; in other words, references from studies identified from primary database search were accurately assessed for further suitable entries that were then included in the subsequent screening stages of the analyses (see further). Only observational studies (i.e., case–control studies [CC], cohort studies [CH], or cross-sectional studies [CS]) were considered suitable for the present systematic review, and suitable articles were included in the analyses if written in any language understood by study authors, i.e., English, Italian, German, French, Spanish, and Portuguese.

In accordance with PRISMA guidelines, suitable articles were screened for their relevance to the subject, initially by title, and, if positively title-screened, by the content of their abstracts [39,40,41]. All articles considered consistent with the aims of the present study were full-text screened by two investigators (I.Z. and S.C.), who assessed the fulfillment of the following inclusion criteria:(1)The study was conducted during or after a documented NiV outbreak;(2)The full text provided the total number of HCWs involved in the management of NiV-positive cases;(3)Detection of Nipah-reactive antibodies documented IgG and/or IgM in serum or CSF samples with and without confirmation tests (e.g., real-time [quantitative] polymerase chain reaction [RT-qPCR]) AND/OR providing the number of HCWs with documented NiV infection).

Studies providing seroprevalence data from less than 25 HCWs were included in the systematic review but arbitrarily removed from the meta-analysis. If the study did not document the occurrence of NiV-reactive antibodies in non-exposed HCWs, the study was still included in the analyses for calculating the detection rate of NiV in occupational settings, but were excluded from analyses for estimating the risk of developing NiV infection in HCWs.

On the contrary, the following exclusion criteria were applied during full-text analysis:(1)the study was a derivative one (i.e., systematic reviews and meta-analyses), letters, editorial comments, and case reports;(2)the study only included data on animals (including non-human primates);(3)the study did not provide details about the geographical setting and corresponding timeframe;(4)the study included the whole of personnel from healthcare facilities hosting NiV patients but did not distinguish between HCWs and professionals not involved in the actual care of patients, such as administrative workers, maintenance workers, etc.

### 2.3. Data Extraction

The following data were extracted from the included studies:(a)Settings of the study (year, month, or season and geographic region);(b)Exposure settings (i.e., including all HCWs from the parent institution(s) or providing data on HCWs exposed vs. not involved in the care for NiV patients);(c)Laboratory testing strategy (i.e., the strategy for identifying NiV-reactive antibodies and corresponding confirmatory tests, if any);(d)Where available, data on the testing of other Henipaviruses (i.e., Hendra virus).

In cases where the cross-posting of results was identified, reports were accurately analyzed to fill the knowledge gaps, therefore providing a more extensive description of cases and eliminating duplicates.

### 2.4. Qualitative Assessment

The quality of studies included in this systematic review was assessed by means of the Newcastle–Ottawa scale (NOS) (potential range: from 0 to 9 points) [42]. The NOS is a review tool for evaluating the risk of bias in observational studies [43]. Due to its easy applications, NOS has been extensively recommended by Cochrane Collaboration for the qualitative assessment of source studies to be implemented into systematic reviews [44]. The NOS scale consists of four domains of risk of bias assessment: (1) selection bias; (2) performance bias; (3) detection bias; and (4) information bias. All of them have been specifically tailored for case–control and cohort studies, while no specific recommendation was originally identified for cross-sectional studies. However, as cross-sectional studies and cohort studies share a similar structure, except for the timing of the measurement for exposures and outcomes (i.e., at the same time, or cross-sectionally, for cross-sectional studies; after the exposure for cohort studies) [45], we opted to apply the cohort studies scale also to cross-sectional ones.

According to the current indications and the study protocols, two investigators independently rated all suitable articles and provided a summary of their potential shortcomings (I.Z., S.C.). Potential disagreements were primarily resolved by consensus between the two reviewers; input from a third investigator (M.R.) was requested and obtained when consensus was not possible.

### 2.5. Data Analysis

#### 2.5.1. Descriptive Analysis

All studies eventually included in the present systematic with meta-analysis were initially summarized in terms of descriptive analysis by calculating corresponding crude prevalence figures for the NiV detection rate. Calculations were distinctively performed by the specimen (e.g., blood/serum and CSF), testing strategy (e.g., IgG vs. IgM), and exposure groups (e.g., HCWs exposed to NiV patients vs. not exposed ones).

#### 2.5.2. Meta-Analysis

Pooled prevalence rates for NiV-targeting antibodies were meta-analyzed through a random effect model (REM) that implemented the inverse variance method and a maximum likelihood estimator for the calculation of tau^2^ (REML). The Freeman Tuckey double arcsine transformation was implemented for the transformation of proportion before meta-analysis and was preferred over logit transformation as it was considered more effective in dealing with samples of reduced and heterogeneous size [46]. Finally, 95%CI of pooled estimates were calculated by means of the highly conservative Clopper Pearson approach [47]. The REM approach was implemented and preferred over the fixed effects model as it is usually considered more effective in dealing with meta-analyses, including a reduced number of sampled studies and high heterogeneity across the source data [48,49]. Similarly, the REML was preferred over other methods (e.g., DerSimonian and Laird) as it excludes the summary effect parameter in its estimation of tau^2^ (unlike to DL), therefore being considered more effective in reducing residual bias [50].

#### 2.5.3. Heterogeneity

Heterogeneity (i.e., the inconsistency of effect among the included studies) was handled by calculating the corresponding I^2^ statistic; that is, the percentage of total variation across the included studies likely occurring because of actual differences rather than chance [51,52]. As suggested by Hippel et al. [52], point estimates of I^2^ statistics calculated from small meta-analyses may provide misleading guidance over the actual heterogeneity. Therefore, the 95%CIs of the point I^2^ estimates were also calculated and reported. In accordance with current recommendations, heterogeneity was considered low for I^2^ values ≤ 25%, moderate for I^2^ values ≥ 26% and <50%, substantial for I^2^ values ranging between 50% and 75%, and considerable for estimates > 75% [44,52,53].

#### 2.5.4. Sensitivity Analysis

Sensitivity analysis can be defined as the study of how the uncertainty in the output of a mathematical model can be apportioned to different sources of uncertainty from the source data of that model. For the aims of the present study, sensitivity analysis for the present meta-analysis was therefore performed by calculating the pooled estimates of prevalence and I^2^ statistics in a model that excluded one study at a time.

#### 2.5.5. Publication Bias

As meta-analyses rely on the quality and reliability of source data, publication bias (i.e., the likelihood of systematic deviation from the truth in the results of a meta-analysis due to the higher likelihood for published studies to be included in meta-analyses than unpublished studies) can lead to misleading recommendations [54]. In order to assess whether collected data were or were not affected by any publication bias, funnel plots were initially calculated. Funnel plots are scatter plots in which the effect estimates from individual studies are plotted on the horizontal axis against the standard error of the estimated effect on the vertical axis. In the presence of publication bias, funnel plots are characterized by an asymmetrical appearance and greater asymmetry will suggest a more significant bias. The asymmetry of funnel plot outcomes with three or more included studies was then assessed using Egger’s test [41,55,56], i.e., a linear regression of the effect estimates on their standard errors weighted by their inverse variance.

A quantitative analysis of publication bias was provided by the calculation of the Luis Furuya-Kanamori (LFK) index and corresponding Doi plots. The LFK index quantifies the skewness of Doi plots, providing a quantitative appraisal of publication bias: LFK values < 1 indicated low or even no asymmetry (i.e., absence of publication bias), LFK values ranging between 1 and 2 hint at minor asymmetry (i.e., moderate publication bias), while LFK values > 2 indicate major asymmetry (i.e., concern for publication bias) [57].

Eventually, small-study bias (i.e., smaller studies showing different, often larger, effects than large ones) was assessed by calculating corresponding radial plots (i.e., a graphical display for comparing estimates that have differing precisions) [58].

A *p*-value < 0.05 was considered statistically significant for both publication and small-study bias.

#### 2.5.6. Software

Mendeley Reference Manager (version 2.121.0; Mendeley Ltd.; New York, NY, USA) was employed to manage suitable articles and perform screening and rating procedures. Calculations required by meta-analysis were performed using R (version 4.4.1) [59] and Rstudio (version 2024.04.2 Build 764; Posit Software, PBC; Boston, MA, USA) software using the packages *meta* (version 7.0), *fmsb* (version 0.7.5), *epiR* (version 2.0.63), and *robvis* (version 0.3.0). Plots were calculated using the R packages *ggplot2* (version 3.4.3), *ggpubr* (version 0.6.0), and GraphPad Prism, version 10.0 (GraphPad Software LLC, Boston, MA, USA).

## 3. Results

### 3.1. Search and Selection Process

A total of 2593 entries were identified from the inquired databases (i.e., 390 from Pubmed, 15.04%; 283 from CINHAL, 10.91%; 966 from EMBASE, 37.25%; 864 from SCOPUS, 33.32%; 90 from MedRxiv, 3.47%; Appendix B Table A1). Overall (Figure 1), 2188 articles were duplicated across the databases, therefore being removed (84.38%). A total of 405 articles were then screened by their titles and abstracts; of them, 285 were removed as they did not fulfill either the research concept or inclusion criteria, or both. The remaining 120 entries were sought for retrieval (4.63%); their full texts were eventually retrieved and individually assessed, leading to the eventual removal of 104 articles as they were not consistent with the inclusion criteria, leaving 16 entries, 0.62% of the original pool (Figure 1) [16,18,19,27,30,31,32,37,60,61,62,63,64,65,66,67]. Regarding their underlying design, the majority of sampled studies had a CS design (10, 62.50%), while four studies had a CH design (25.0%), and two studies a CC design (12.50%). Overall, six studies provided data on the seroprevalence of IgM/IgG against NiV [19,27,37,61,62,63], eight studies reported on the outcome of occupationally acquired HCWs infections irrespective of their serology [16,18,30,31,32,64,65,66,67], and one study provided data on both the serology and outcome of sampled individuals [60]. Focusing on seroprevalence studies, two of them were based on enzyme immunoassay [27,60], while five studies included data based on ELISA [19,37,61,62,63]; interestingly, both EIA studies and three out of the five ELISA-based studies relied on United States CDC-provided antigens [19,27,61,62], while the more recent studies from Ramachandran et al. [63] and Yadav et al. [37] deliberately implemented indigenous-derived antigens, with high sensitivity and specificity compared to the CDC-provided ones (i.e., 100% and 83.3%, respectively) [68].

### 3.2. Summary of Included Studies

Data on seroprevalence studies are reported in Table 1, while data from studies providing the outcome of sampled HCW infections are summarized in Table 2.

Overall, available studies included data from December 1998 [60] to August 2021 [37] from four geographic areas, namely Malaysia [60], Singapore [61], Philippines [16], and the Indian subcontinent, including both Indian states of Bengal [30] and Kerala [19,31,32,37,63,66,67], and Bangladesh [27,62,64,65]. Their geographic distribution alongside that of the host fruit bats from the *Pteropus* genus are shown in Figure 2.

### 3.3. Risk of Bias Analysis

A summary of the quality assessment is provided in Table 3. In summary, the quality of most of the included studies was labelled as high or even very high, as 14 out of the 16 studies had a cumulative score equal to or higher than 7 out of 9.

Only the studies from Ching et al. [16] and Gurley et al. [62] were affected by several shortcomings in various domains, most notably when dealing with the representativity of the sample and the ascertainment of the baseline condition. However, despite the good quality of the reports, the remaining studies were affected by some potential issues. For instance, the study from Mounts et al. [60] did not provide a clear explanation for the recruitment of sampled HCWs from the original series of occupationally exposed workers. Another possibly biased study, at least from the aims of the present report, was that of Ramachandran et al. [63], as it was initially designed to provide a description of individuals possibly exposed to an index case, including the 26 sampled HCWs. In fact, some degree of bias, in terms of selection bias, similarly affected other reports on NiV outbreaks and, more precisely, the studies from Arunkumar et al. [32], Chakraborty et al. [64], and Chadha et al. [30], as their design did not guarantee that all potentially exposed HCWs were ultimately included in the study. Similarly, neither the study from Chadha et al. [30] and Nikolay et al. [65] provided accurate information on the outcome of sampled HCWs. Nonetheless, it is otherwise important to stress that the report from Nikolay et al. [65], similarly to the study from Ramachandran et al. [63], was not designed for an accurate description of NiV cases in healthcare settings, and these characteristics potentially affected data reporting.

### 3.4. Main Findings

Overall, seroprevalence studies included a total of 1300 HCWs, but seroprevalence data were provided by a total of 910 subjects (67.75% of potentially sampled HCWs), with a total of one positive case for IgM and five positive cases for IgG, for crude prevalence estimates of 0.11% and 0.55%, respectively. In the end, only two studies [27,60] provided prevalence estimates for IgG/IgM targeting NiV not only in HCWs involved in the management of index cases, but also in professionals not likely exposed, for a total of 302 professionals. In fact, no positive cases were identified either for IgM or for IgG class NiV-targeting antibodies.

Studies on the outcome of NiV infections included data on a total of 740 NiV cases; of them, 45 occurred in HCWs (6.08%). As the studies from Nikolay et al. [65] and Chadha et al. [30] did not provide the actual number of NiV-associated deaths, CFR estimates were calculated on a total of 17 HCWs, with a total of 11 deaths (crude CFR 64.71%).

### 3.5. Meta-Analysis of Prevalence Estimates

The results of the REM meta-analysis are reported in Figure 3. Briefly, pooled prevalence for IgM class antibodies was estimated at 0.00% (95%CI 0.00 to 0.10). Heterogeneity was seemingly low in terms of point estimates (I^2^ = 0.0%; tau^2^ = 0.0; Q = 2.78, *p* = 0.835), while the calculation of corresponding 95%CI of I^2^ statistics rather hinted at a substantial heterogeneity (95%CI 0.0 to 70.8) (Figure 3a). Subgroup analyses were affected by the single positive case identified, with prevalence estimates of 0.07% (95%CI 0.00 to 0.71; I^2^ = 0.0%) for ELISA-based studies including indigenous antigens, compared to 0.00% (95%CI 0.00 to 0.13; I^2^ = 0.0%) for studies based on EIA and 0.00% (95%CI 0.00 to 2.70; I^2^ = 0.0%) for studies based on ELISA with CDC-provided antigens, but the differences were not significant (Q = 0.80, *p* = 0.671).

Pooled prevalence for IgG class antibodies (Figure 3b) was otherwise estimated at 0.08%, 95%CI 0.00 to 0.72, with a point estimate of I^2^ statistics of 1.1% (tau^2^ = 0.001, Q = 6.07, *p* = 0.416). The corresponding 95%CI of I^2^ statistics otherwise hinted at a substantial heterogeneity (0.0 to 71.1%). In subgroup analyses, the highest estimates were associated with EIA studies (0.38%, 95%CI 0.00 to 1.63), followed by ELISA studies relying on CDC-provided antigens (0.20%, 0.00 to 1.60), and ELISA studies based on indigenous antigens (0.00%, 95%CI 0.00 to 2.70), but again the differences were not significant (Q = 0.59, *p* = 0.746).

### 3.6. Meta-Analysis of CFR

Overall (Appendix B, Figure A1), the proportion of HCWs among the sampled individuals with a history of NiV infection was 17.65% (0.84 to 44.33), with considerable heterogeneity across sampled studies (I^2^ = 88.7%, *p* < 0.001, tau^2^ 0.05, 95%CI 0.00 to 0.41). In the REM meta-analysis on the CFR among HCWs with a documented NiV infection, a pooled estimate of 73.52% (95%CI 34.01 to 99.74) was calculated, with moderate to substantial heterogeneity (I^2^ = 29.6%, 95%CI 0.0 to 68.5%) (Figure 4).

Subgroup analysis was performed by reference country; the highest CFR was associated with cases from Bangladesh (two studies, CFR 100%, 95%CI 21.26 to 100, I^2^ = 0.0%), followed by Indian cases from Kerala (four studies, CFR 75.80%, 95%CI 36.38 to 99.99, I^2^ = 0.0%). The single studies from Philippines (CFR 100%, 95%CI 30.27 to 100) and Malaysia (CFR 0.00%, 95%CI 0.00 to 50.03) were considered in the subgroup comparisons, which identified significant differences between groups (Q = 8.17, *p* = 0.043).

### 3.7. Publication Bias

Publication bias was initially ascertained by calculating corresponding funnel plots for seroprevalence estimates of IgM and IgG (Figure 5a,b, respectively) and of the CFR (Figure 5c). According to their design, the sample size was plotted against the effect size (i.e., detection rate and CFR) in the funnel plots. As the size of the sample increased, individual estimates of the effect converged around the true underlying estimate [54].

Briefly, as somehow expected due to the uneven sample size of all estimates, all funnel plots were asymmetrical, suggesting the likelihood that pooled data were affected by a substantial publication bias. However, as shown in Figure 6, the calculation of Doi plots and corresponding LFK indexes hinted at the presence of some degree of publication bias only for seroprevalence estimates for IgM-class antibodies (LKF index = 5.30), while estimates for IgG-class antibodies suggested low risk for publication bias (LFK index = −0.30), and estimates for the CFR were reasonably affected by moderate publication bias (LFK index = 1.14).

In fact, as shown in Table 4, all results of Egger’s test were associated with a *p* value > 0.05, which seemingly ruled out the underlying publication bias. Consequently, reported asymmetry must be explained by other potential sources of bias, including the heterogeneous choices in the outcome measure and sampling strategies, but also in the different laboratory options (i.e., EIA vs. ELISA, but also ELISA with CDC-provided antigens vs. indigenous antigens).

Eventually, radial plots (scatter plots of standardized estimates) were calculated for studying the corresponding small-study bias (Figure 5d–f). Plots calculated for the detection rates of IgM (Figure 5d) and IgG (Figure 5e) and also of the CFR (Figure 5f) were characterized by an uneven distribution of the point estimates across the regression lines, suggesting that the estimates may have been somehow affected by smaller samples. On the contrary, estimates of the CFR were unevenly scattered across the regression line being this remark, and this is consistent with small-study bias.

### 3.8. Sensitivity Analysis

The results of the sensitivity analysis are summarized in Figure 7. Even though only the study from Kumar et al. [19] reported on positive cases for IgM-class antibodies (Figure 7a), its removal did not substantially impact the overall estimates. When dealing with IgG-class NiV-targeting antibodies (Figure 7b), the removal of the studies from Mounts et al. [60] and Gurley et al. [62] led to a substantial fall of estimates for detection rates (0.00%, 95%CI 0.00 to 0.62 and 0.00%, 95%CI 0.00 to 0.45, respectively), while the removal of the study from Chan et al. [61] increased the detection rate to 0.21% (95%CI 0.00 to 0.95).

Regarding the CFR (Figure 7c), the removal of Chandni et al. [31] and Ching et al. [16] resulted in a reduction in corresponding estimates to 67.28% (95%CI 24.87 to 98.91) in both cases, while the removal of Mounts et al. [60] resulted in an increase in the CFR to 87.01% (95%CI 54.46 to 100).

## 4. Discussion

### 4.1. Summary of Main Results

In this systematic review and meta-analysis, we were able to retrieve a total of seven seroprevalence studies on the seroprevalence of NiV among HCWs [19,27,37,60,61,62,63]; of them, the study from Mounts et al. [60] also provided data on the outcome of included cases. Nine further studies reported on the documented NiV infections among HCWs exposed to index cases [16,18,30,31,32,64,65,66,67]. As two studies did not provide the eventual number of fatalities among the infected HCWs [30,65], estimates on the CFR were calculated on a reduced pool of cases (i.e., eight studies and seventeen HCWs) [16,18,31,32,64,66,67]. A quantitative summary by means of REM meta-analysis calculated seroprevalence rates of 0.00% (95%CI 0.00 to 0.10) for IgM-class antibodies, compared to 0.08% (95%CI 0.00 to 0.72) for IgG class-antibodies, but it is important to stress that in four out of seven sampled studies, no seropositive cases were identified among HCWs [27,37,61,63]. The CFR was estimated at 73.52% (95%CI 34.01 to 99.74), but again, this estimate is reasonably affected by the two largest studies, i.e., Chadha et al. [30] and Nikolay et al. [65], having not provided the actual outcome of sampled HCWs.

### 4.2. Generalizability of the Results

Our meta-analysis showed that NiV has a considerable epidemic or pandemic potential, with a very high CFR, usually ranging between 40 and 75% or even higher, depending on the NiV strain, access and quality of healthcare, and underlying conditions [4,5,26,69,70]. On the one hand, the present study confirmed that work-related NiV infections among HCWs involved in the management of index cases are associated with a very high CFR, with 11 deaths out of 17 HCWs with a documented NiV infection [16,18,30,31,32,64,65,66]. On the other hand, our results suggest that NiV may be quite ineffective in achieving inter-human transmission in healthcare settings; in fact, among the sampled HCWs, a seropositive status was only documented in one case for IgM-class antibodies [20] and in five cases for IgG-class antibodies [60,62]. As both IgG- and IgM-class antibodies are considered to persist in the blood of NiV-infected patients for several months, and all included studies were performed in a timely manner, it is quite unlikely that pooled results may have been affected by some degree of underestimation [62,63]. This is particularly significant from a public health point of view and from an occupational health point of view, as all the included seroprevalence studies were performed before the SARS-CoV-2 pandemic. In other words, all reported outbreaks were documented in healthcare settings where infection prevention and control practices were still not extensively implemented in terms of personal protective equipment (PPE), enhanced contact precautions, and the emphasis on hand hygiene [71]. For example, the detailed report from Gurley et al. [62] documented a very low seroprevalence rate among 105 HCWs involved in the management of NiV infections in a Bangladesh hospital despite the inconsistent use of PPE while handling NiV-infected patients.

The relatively low risk for NiV infection among HCWs should not divert the attention required to properly cope with a pathogen that cannot be either prevented by effective vaccines or managed by specific therapies. On the one hand, several NiV-candidate vaccines are currently being evaluated in clinical trials [72,73], including an mRNA vaccine (mRNA-1215, encoding a chimeric antigen based on the prefusion F glycoprotein and the G glycoprotein) [74,75], an innovative vaccine targeting CD40 receptor of antigen-presenting cells by fusing a humanized anti-CD40 monoclonal antibody to the NiV G protein (CD40.NiV) [73], and a recombinant chimpanzee adenovirus-based vaccine (AdC68-G) sharing the blueprint of COVID-19 vaccine ChAdOx1 nCoV-19, the Ad26.COV2.S COVID-19 vaccine, and the respiratory syncytial virus Ad26.RSV.preF vaccine [76,77,78], expressing the codon-optimized full-length NiV G protein (ChAdOx1 NipahB vaccine) [79]. Notably, the first human clinical trial of an NiV vaccine rolled out in the first months of 2024 through the delivery of the ChAdOx1 NipahB vaccine in adults aged 18 to 55 years; as of December 2024, the recruitment has halted, but first results are still anticipated [80]. On the other hand, medical research on specific therapies faces several difficulties, most notably the reduced timeframe of reported outbreaks. As most recorded outbreaks, particularly from Bangladesh and Kerala, usually last a few days, drug trials are quite challenging to design and implement. In this regard, the study from Chandni et al. [31] provided some interesting insights, as authors reported several non-specific antiviral drugs delivered preventing other viral etiologies. While acyclovir and oseltamivir did not alter the course of infections, ribavirin seemingly reduced the CFR by around 20% in the later stages of the outbreak, a result considered consistent with a previous study from Malaysia [81]. Monoclonal antibodies (mAbs) have been increasingly acknowledged as a reliable option in facing infectious diseases [82,83], as stressed by the delivery of palivizumab and nirsevimab for preventing respiratory syncytial virus infections [84,85], and several mAbs targeting NiV have been recently considered for prevention and treatment [75,86,87]. For example, the human cross-reactive mAb 102.4 neutralizes NiV attachment glycoprotein G, being quite effective in avoiding host infection in both ferret and non-human primates [87], but also in avoiding neurological complications when delivered after the infection and even after the onset of the viremia.

### 4.3. Implications for Daily Practice

During the first documented NiV outbreaks, including the first occupational health studies, in most cases, episodes of the interhuman transmission of this pathogen were not documented, and nearly all infections were associated with the exposure of human cases to affected animals or fruits and vegetables contaminated with animal fluids [2,5,6,7,8,9,12,13,14]. On the contrary, outbreaks in the Indian subcontinent were characterized by documented inter-human transmission [18,21,32,35,66,88,89,90], stressing the pandemic potential that has been allegedly associated with NiV, particularly after the inception of the SARS-CoV-2 pandemic [36]. Even though NiV still appears to be confined to areas corresponding with the documented habitats of the main host (i.e., bats in the *Pteropus* genus), the inter-human transmission of NiV has resulted in significant implications for all professionals caring for affected patients. NiV has not only emerged as a potential occupational risk agent [28,62], but also as a nosocomial one, affecting HCWs as well as patients and individuals visiting and/or assisting friends and relatives during their institutionalization [60,62,91]. Following the blueprint learned during the SARS-CoV-2 and mpox pandemic [34,92,93], but also when dealing with more “conventional” viral pathogens such as varicella-zoster virus and measles [94,95,96,97,98], designing, tailoring, and properly implementing appropriate preventative occupational health policies therefore represent a very effective way for simultaneously achieving HCWs’ and patient safety. Occupational physicians (OP), who are the medical professionals responsible for health prevention and promotion in the workplaces, actively contribute to the prevention of biological risk agents by applying and tailoring official recommendations [99,100], by either providing or promoting appropriate immunizations [100,101,102,103,104], by contributing to the identification and promotion of appropriate PPE [34], and by specifically tailored post-exposure prophylaxis procedures [105].

Interestingly enough, implementing the face mask mandate in patients, caregivers, and HCWs and avoiding unnecessary interactions between patients, their caregivers, and healthcare personnel have been identified as quite effective measures for countering the spreading of NiV in enclosed hospital settings [21].

Far from sharing a potentially deleterious false sense of security, our results suggest that by applying rigorous infection prevention and control practices, even in healthcare settings characterized by a reduced availability of medical resources, all HCWs (i.e., all medical professionals, nurses, and allied medical professionals) could safely perform their daily duties and care for NiV-infected patients. Interestingly, the effective prevention of NiV was achieved even before the COVID-19 pandemic raised the level of attention paid to infection control protocols, stressing that NiV can be controlled in a very cost-effective way [21,90].

Nonetheless, interhuman spreading and the subsequent epidemic and even pandemic potential of NiV should not be underestimated. For instance, data from previous outbreaks hint that NiV could quite effectively spread in hospital settings where index cases were not properly isolated, affecting all bystanders not benefiting from infection control protocols, mask-wearing policies, and proper room ventilation [62,106,107]. In this regard, the study from Gurley et al. [62] provided further significant insight. While no one among the 105 sampled HCWs exhibited IgM-class antibodies, two nursing students who reported having changed bed sheets for index cases had IgG-class antibodies against NiV; even though neither student reported any unprotected exposure to bodily fluids of inpatients with documented NiV infection, their use of PPE (particularly masks and gloves) was not seemingly accurate.

### 4.4. Limitations

Several potential shortcomings of the present systematic review with meta-analysis must be addressed, particularly the very high heterogeneity of the sampled studies, which can be explained by the individual characteristics of source data.

In the first place, the quality of the evidence provided by a systematic review with meta-analysis depends on that of the parent studies [39,40,41]. In the risk of bias analysis performed by means of NOS scale, the quality of the source studies was quite uneven, which in turn likely depends on the limited availability of source cases. Since the original identification of NiV as a suitable zoonotic pathogen [13,25], NiV outbreaks were characterized by the sudden onset and limited timeframe. Not only have temporospatial constraints impaired the proper design and conduction of interventional studies, but observational studies may also be biased due to the inappropriate recall of critical information, particularly during the early stage of the outbreak, leading to the failure to properly identify the episodes associated with the potential occupational exposure of HCWs, ultimately impairing the dichotomization between likely/certainly exposed vs. not likely/certainly exposed HCWs [15,28,90,108,109].

Another potential source of bias is represented by the limited viability of IgM-class specific antibodies and the potential cross-reactivity with other members of the *Henipavirus* genus, i.e., Hendra virus [11,25,75]. Even though it is quite unlikely that during the outbreaks included in the present summary of evidence, any co-circulation of Hendra virus may have occurred, the need for laboratory testing that is able to distinguish between NiV and Hendra virus may have limited the diagnostic opportunities. Notably, laboratory techniques only focused on the serology of NiV infection [60,61]: while it is quite unlikely that NiV-targeting IgM-class antibodies may be detectable after the fourth post-exposure month, serology cannot specifically identify the timing of the original infection [63]. In other words, among the sampled HCWs, the effective work-related NiV infection following the workers’ exposure to documented NiV cases should be acknowledged as a suitable option. Nonetheless, due to the current epidemiology of NiV, even in settings such as Kerala and Bengal, the nosocomial transmission of the pathogen remains the most likely option [18,21,22,31,32,35,66,88,89,90].

Thirdly, as the studies span the decades between 1998 and 2019, diagnostic strategies have evolved over time, providing increasingly accurate and effective diagnostic options, even in the limited-resource areas that have been particularly affected by the outbreaks included in the present systematic review with meta-analysis [4,71]. The very same reference antigens evolved over time, as original specimens provided by United States CDC and derived from the original outbreaks from Malaysia and Singapore (i.e., NiV-A) have been replaced by indigenous-derived antigens from NiV-B, with allegedly improved diagnostic performances [37,63,68], but also potential constraints when pooling and comparing corresponding seroprevalence data.

Fourthly, we must stress the potentially limited representativity of the pooled sample. Seroprevalence studies gathered data on a total of 910 HCWs, while the total number of exposed professionals reported by studies on NiV outcomes was not actually provided by all parent studies. While the studies from Mounts et al. [60], Chan et al. [61], Gurley et al. [62], and Kumar et al. [19] reasonably provided a comprehensive report on the HCWs exposed to NiV during the documented outbreaks, the studies from Hsu et al. [27], Ramachandran et al. [63], and Sazzad et al. [67] included a more restricted sample whose definition was more unclear. For example, the latter study includes a total of 328 cases, representing around 44.32% of the total NiV cases included into the pooled analyses on the outcome of viral infection. As the status of HCWs was undoubtedly associated with a single case—a physician who developed a fatal infection—while whether other HCWs were actually included into the sample remains undisclosed, a single HCW from this study was added to the pooled sample. Consequently, we cannot rule out that the very CFR we documented may be due to the oversampling of severe cases. In this regard, a significant proportion of seroprevalence studies on NiV cases do not provide the total number of involved and/or occupationally exposed HCWs, rather focusing on subjects having been exposed to bodily fluids and respiratory specimens of affected cases, a definition that therefore does not specifically refer to HCWs but that can include all individuals assisting NiV cases during their hospitalization [110,111,112,113,114].

Fifthly, studies were quite heterogeneous in terms of their design, and the meta-analysis included two case–control study, four cohort studies, and ten cross-sectional studies. Cohort studies and cross-sectional studies are usually considered comparable in design, with the notable difference being that in the cross-sectional design, exposures and outcomes are measured at the same time (i.e., cross-sectionally), whereas in a cohort study, outcomes are typically measured after the exposure has been measured (i.e., longitudinally) [45]. Due to the characteristics of immune responses to NiV and to the dubious timing of individual exposure to NiV, the different underlying design has possibly led to some degree of eventual inaccuracy in the pooled estimates.

Sixthly, retrieved studies were reasonably affected by a substantial small-study effect. In fact, the analyses of the radial plot stress a residual small-study effect, as confirmed by the detailed appraisal of individual studies, particularly for the pooling of CFR estimates, as the outcome of only seventeen cases of NiV infections was provided, including five studies encompassing a total of seven cases which reported a CFR of 100% [16,31,64,66]. Therefore, we cannot rule out that the reports above may have led to larger effect estimates than the remaining ones, with an eventual overestimation of the actual lethality of NiV among HCWs.

## 5. Conclusions

Our systematic review with meta-analysis has documented how NiV may result as a possible occupational infection among HCWs involved in managing incident cases. Even though the high CFR among HCWs may result from sampling and reporting issues, NiV can lead to a severe clinical syndrome associated with high rates of complications, often fatal, and preventive interventions should be carefully implemented to protect healthcare professionals. On the other hand, as most NiV outbreaks occurred in limited-resource settings, a reasonable conclusion is that basic preventive measures, including the use of PPE and the appropriate isolation of incident cases with physical distancing, may be quite effective in avoiding the occurrence of new infections among HCWs.

## Figures and Tables

**Figure 1 viruses-17-00081-f001:**
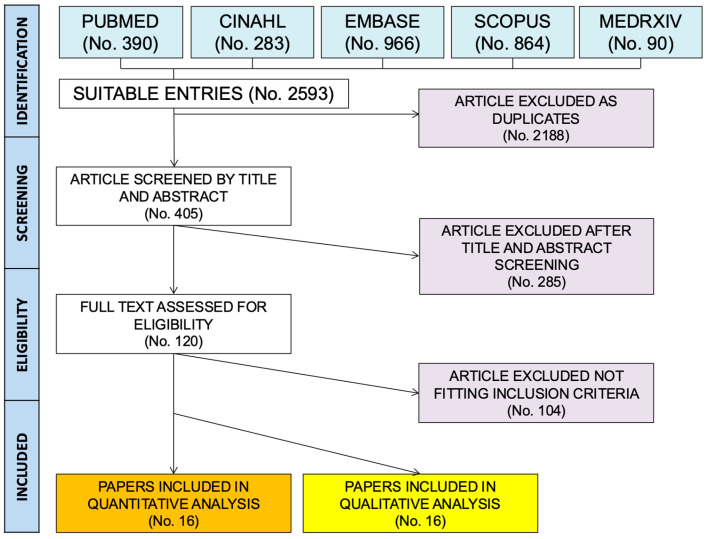
Flow chart of included studies [16,18,19,27,30,31,32,37,60,61,62,63,64,65,66,67].

**Figure 2 viruses-17-00081-f002:**
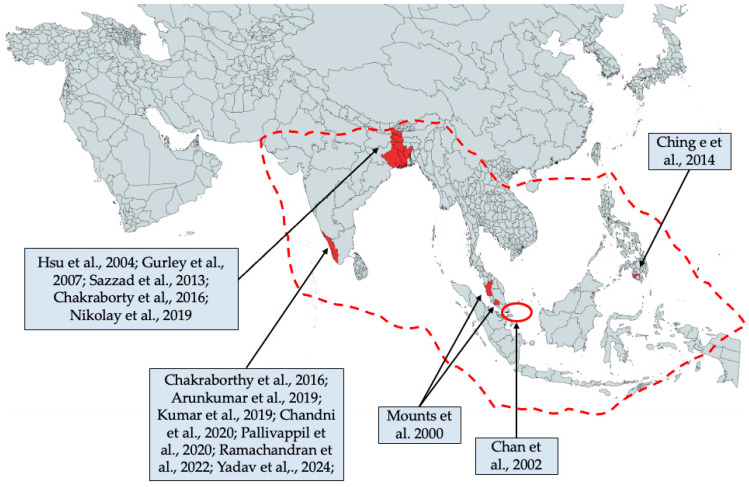
Geographic distribution of retrieved studies. Dotted line represents the geographic distribution of the main documented hosts of Nipah virus, the fruit bats or flying foxes from *Pteropus genus*. Numbers between brackets are the reference to the retrieved studies [16,18,19,27,30,31,32,37,60,61,62,63,64,65,66,67]. The present map has been created by means of mapchart.net, licensed under a Creative Commons Attribution-ShareAlike 4.0 International License.

**Figure 3 viruses-17-00081-f003:**
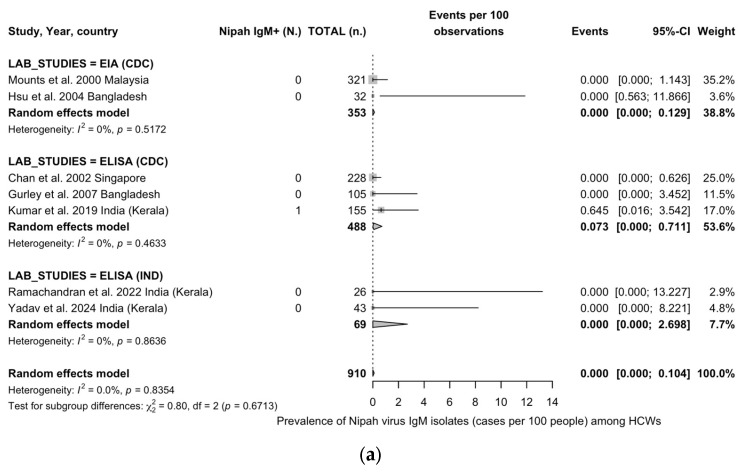
Forest plots for the prevalence of Nipah virus antibodies in healthcare workers (HCWs). Estimates for IgM class antibodies are provided in subfigure (**a**). Estimates for IgG class antibodies are provided in subfigure (**b**) [19,27,37,60,61,62,63] (Note: 95%CI = 95% confidence interval; EIA = enzyme immunoassay; ELISA = enzyme-linked immunosorbent assay).

**Figure 4 viruses-17-00081-f004:**
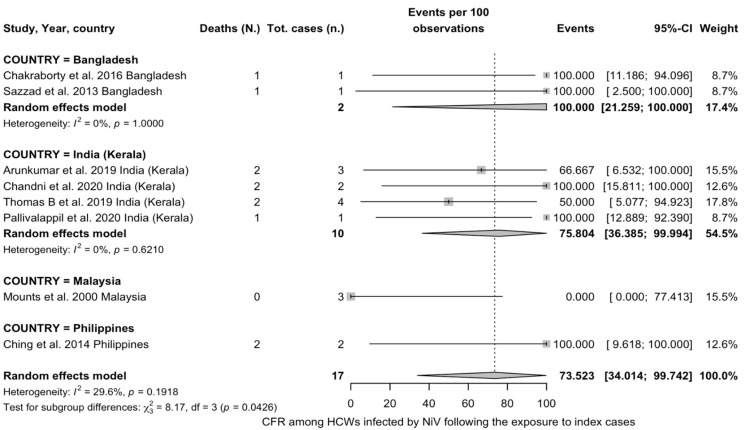
Forest plots for case fatality ratio (CFR) among healthcare workers (HCWs) infected by Nipah virus (NiV) following their exposure to index cases [16,18,30,31,32,60,64,65,66,67].

**Figure 5 viruses-17-00081-f005:**
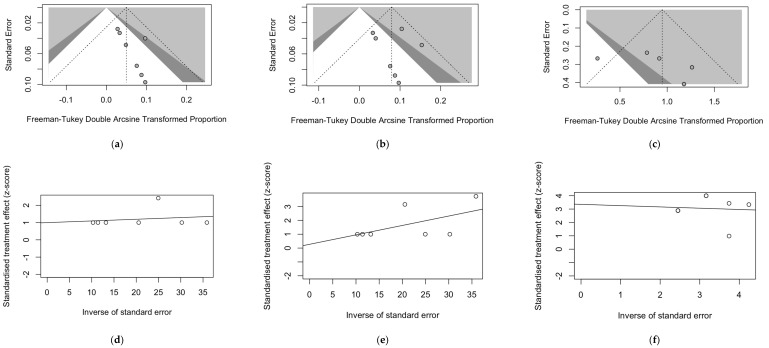
Funnel plots for the prevalence of IgM (**a**) and IgG (**b**) against Nipah (NiV) virus and for the reported case fatality ratio for NiV-related infections from retrieved studies (**c**). Similarly, subfigures (**d**–**f**) provide the radial plots on the publication bias for the reported prevalence of NiV-targeting IgM and IgG and for the NiV-related case fatality ratio.

**Figure 6 viruses-17-00081-f006:**
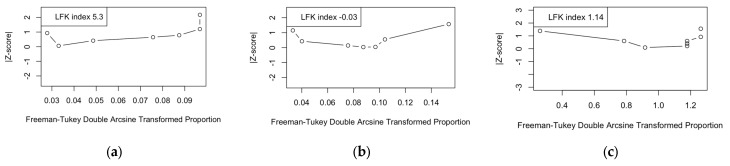
Doi plots for estimates on seroprevalence of IgG (**a**) and IgM (**b**) class antibodies and for case fatality ratio (**c**). For each plot, corresponding Luis Furuya-Kanamori (LFK) index is reported. Values exceeding 2 were major asymmetry and suspected publication bias.

**Figure 7 viruses-17-00081-f007:**
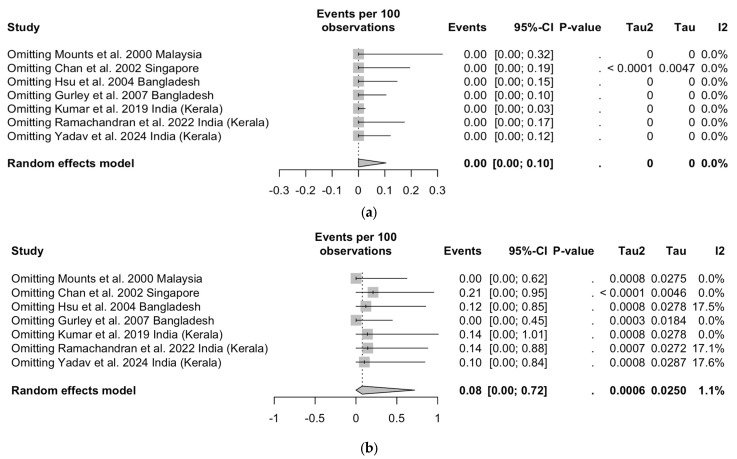
Forest plots reporting on the sensitivity analysis for the seroprevalence of Nipah virus (NiV) targeting IgM (**a**) and IgG (**b**) antibodies and on the NiV-related case fatality ratio (**c**) in healthcare workers [16,18,19,27,30,31,32,37,60,61,62,63,64,65,66,67].

**Table 1 viruses-17-00081-t001:** Summary of retrieved seroprevalence studies (Note: HCWs = healthcare workers; EIA = enzyme immunoassay; ELISA = enzyme-linked immunosorbent assay; CH = cohort study; CS = cross-sectional study; IND = indigenous reagents; CDC = United States Centers for Diseases control and prevention-provided reagents) [19,27,37,61,62,63].

Study	Country	Timeframe	Laboratory	Design	Total HCWs (N/1343)	Sampled HCWs (N/Total HCWs, %)	Positive Cases (N/sampled HCWs, %)		Deaths (N/Sampled HCWs, %)	Data on Not-Exposed HCWs		
							(IgG)	(IgM)		Total (N)	Positive IgG	Positive IgM
Mounts et al., 2000 [60]	Malaysia	26 December 1998 30 April 1999	Serum EIA (CDC)	CH	651 (48.47%)	321 (49.31%)	3 (0.93%)	0 (-)	0 (-)	288	0 (-)	0 (-)
Chan et al., 2002 [61]	Singapore	March 1999	Serum ELISA (CDC)	CH	228 (16.98%)	228 (100%)	0 (-)	0 (-)	0 (-)	-	-	-
Hsu et al., 2004 [27]	Bangladesh	20 April 2001 20 May 2001	Serum EIA (CDC)	CS	32 (2.38%)	32 (100%)	0 (-)	0 (-)	0 (-)	14	0 (-)	0 (-)
Gurley et al., 2007 [62]	Bangladesh	24 March 2004 30 March 2004	Serum ELISA (CDC)	CS	105 (7.81%)	105 (100%)	2 (1.90%)	0 (-)	0 (-)	-	-	-
Kumar et al., 2019 [19]	India (Kerala)	May 2018	Serum ELISA (CDC)	CS	235 (17.50%)	155 (65.96%)	0 (-)	1 (0.65%)	0 (-)	-	-	-
Ramachandran et al., 2022 [63]	India (Kerala)	June 2019	Serum ELISA (IND)	CS	49 (3.65%)	26 (53.06%)	0 (-)	0 (-)	0 (-)	-	-	-
Yadav et al., 2024 [37]	India (Kerala)	August 2021	Serum ELISA (IND)	CH	43 (3.20%)	43 (100%)	0 (-)	0 (-)	0 (-)	-	-	-

**Table 2 viruses-17-00081-t002:** Summary of studies reporting on Nipah virus associated infections among healthcare workers (Note: HCWs = healthcare workers; CFR = case fatality ratio; CC = case–control; CH = cohort; CS = cross-sectional; n.a. = not available) [16,18,30,31,32,60,64,65,66,67].

Study	Country	Timeframe	Design	Total Cases (N/740)	HCWs (n/N, %)	Deaths (N)	CFR (%)
Mounts et al., 2000 [60]	Malaysia	26 December 1998 30 April 1999	CH	3 (0.41%)	3 (100%)	0	0
Chadha et al., 2006 [30]	India (Siliguri)	2001	CS	44 (5.95%)	25 (56.82%)	n.a.	n.a.
Sazzad et al., 2013 [67]	Bangladesh	January 2010 April 2010	CC	328 (44.32%)	1 (0.30%)	1	100%
Ching et al., 2014 [16]	Philippines	April 2014 May 2014	CH	11 (1.49%)	2 (18.18%)	2	100%
Chakraborty et al., 2016 [64]	Bangladesh	23 December 2010 1 March 2011	CC	43 (5.81%)	1 (2.33%)	1	100%
Thomas et al., 2019 [18]	India (Kerala)	April 2018 May 2018	CS	5 (0.68%)	4 (80.00%)	2	50.00%
Nikolay et al., 2019 [65]	Bangladesh	April 2001 April 2014	CS	248 (33.51%)	3 (1.21%)	n.a.	n.a.
Arunkumar et al., 2019 [32]	India (Kerala)	2018	CS	23 (3.11%)	3 (13.04%)	2	66.67%
Chandni et al., 2020 [31]	India (Kerala)	May 2018	CS	12 (1.62%)	2 (16.67%)	2	100%
Pallivalappil et al., 2020 [66]	India (Kerala)	May 2018	CS	23 (3.11%)	1 (4.35%)	1	100%

**Table 3 viruses-17-00081-t003:** A summary of the risk of bias assessment according to the Newcastle–Ottawa quality assessment scale (NOS) for cohort studies and case–control studies. As no scale for cross-sectional studies has been originally developed, cross-sectional studies included in the present systematic review were evaluated by means of the scale for cohort studies (* = item adequacy to the NOS scale; A maximum of two stars can be given for Comparability for confounders in cohort studies and ascertainment of exposure for case-control studies).

Cohort Studies	Selection			Comparability		Outcome			Total
	Representativeness	Exposed Cohort	Ascertainment	Result Not Present at the Start of the Study	Comparability for Confounders	Assessment of the Outcome	Follow-Up Duration	Adequacy of Follow-Up	N/9
Mounts et al., 2000 [60]		*	*	*	**	*	*	*	8/9
Chadha et al., 2006 [30]		*	*	*	**	*	*		7/9
Ching et al., 2014 [16]		*	*	*		*	*	*	6/9
Thomas et al., 2019 [18]	*	*	*	*	*	*	*	*	8/9
Nikolay et al., 2019 [65]		*	*	*	**	*	*		7/9
Arunkumar et al., 2019 [32]		*	*		**	*	*	*	7/9
Chandni et al., 2020 [31]		*	*		**	*	*	*	7/9
Pallivalappil et al., 2020 [66]	*	*	*		**	*	*	*	8/9
Chan et al., 2002 [61]	*	*	*		*	*	*	*	7/9
Hsu et al., 2004 [27]	*	*	*		*	*	*	*	7/9
Gurley et al., 2007 [62]	*	*	*			*	*	*	6/9
Kumar et al., 2019 [19]	*	*	*	*	**	*	*	*	9/9
Ramachandran et al., 2022 [63]		*	*		**	*	*	*	7/9
Yadav et al., 2024 [37]	*	*	*	*	**	*	*	*	9/9
**Case Control Studies**	**Selection**				**Comparability**		**Exposure**		
	Case definition	Representativeness (cases)	Selection of controls	Definition of controls	Study controls comparable for the assessed factor	Study controls comparable for additional factors	Ascertainment	Non-response rate	
Chakraborty et al., 2016 [64]	*			*	*	*	**	*	7/9
Sazzad et al., 2013 [67]	*	*	*	*	*	*	**	*	9/9

**Table 4 viruses-17-00081-t004:** Summary of Egger’s test results on main findings reported in this meta-analysis on Nipah virus (NiV) seroprevalence among healthcare workers and NiV-associated case fatality ratio.

Finding	t	df	Bias (SE)	tau^2^	*p*-Value
Seroprevalence, IgM	1.84	5	0.997 (0.543)	0.333	0.126
Seroprevalence, IgG	0.27	5	0.273 (1.030)	1.197	0.801
Case fatality ratio for NiV infections	1.85	6	3.354 (1.817)	1.057	0.114

## Data Availability

Data are available on request to the corresponding author.

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
