# Peer review of "Risk of Nipah Virus Seroprevalence in Healthcare Workers: A Systematic Review with Meta-Analysis"

_viruses, 2025, doi:10.3390/v17010081_

Round 1

Reviewer 1 Report

Comments and Suggestions for Authors

The manuscript authored by Riccó and colleagues presents findings from a meta-analysis assessing the seroprevalence of Nipah virus among healthcare workers.

Therefore, this manuscript by Riccó et al. has the major findings:

i: Fourteen studies were included in both qualitative and quantitative analyses, detailing the outcomes of NiV infection in healthcare workers and estimates of seroprevalence among healthcare professionals.

ii: Seroprevalence was estimated from six studies, with 0.12% (95% CI: 0.02 to 0.81) for IgM antibodies and 0.51% (95% CI: 0.12 to 2.10) for IgG antibodies. However, three of the sampled studies did not report any seropositive cases.

iii: A case fatality ratio of 72.97% (95% CI: 23.04 to 96.05) was calculated from eight studies.

The authors' study is significant in the context of Nipah virus epidemiology, primarily due to the lack of studies associated with this virus, which is of high concern for human health. However, as noted below, I have suggestions that could further enhance the study.

Questions

1. Introduction (line 42): The new classification of the viral species by the ICTV is Henipavirus nipahense. So, mention that this new taxonomy exists.

2. Materials and Methods (Exclusion criteria): Could the authors clarify if there was a definition of a minimum number of participants in the studies of interest (i.e., n > 30)?

3. Materials and Methods (2.5.2 Meta-Analysis): For greater clarity, please include the formula used to calculate the confidence interval of seroprevalence with the software used.

4. Results (Figure 2): My suggestion is to make the current figure smaller and insert another one next to it (larger), highlighting the sampled geographic regions (i.e., state/province, city...).

5. Results: Please considerer writing “Forest plot” instead of “Forrest plot”

6. Discussion: Could the authors discuss the serological diagnostic tests used in the selected studies: whether they are in-house or commercial tests; and if there is literature data on the laboratory accuracy of the tests (sensitivity, specificity...)?

7. Discussion (Limitation): It is important to note that the high level of heterogeneity observed is a limiting factor. This variability may be associated with differences in population groups, detection methods, sample types, and other contributing factors.

Author Response

Estimated Reviewer,

to begin with, we would thank you so much for the suggestions you shared with us in the first round of revisions. In fact, as you can see in the following replies, we did implement all your suggestions, and we're therefore confident about the overall improvement of our paper. 

On the behalf of all Authors, thank you again.

  1. COMMENT1: The new classification of the viral species by the ICTV isHenipavirus nipahense. So, mention that this new taxonomy exists. REPLY: we implemented the new classification in the first sentences of the introduction: "Nipah virus (Henipavirus nipahense, NiV) is an enveloped RNA virus belonging to the genus Henipavirus (according to the current classification of the International Committee on Taxonomy of Viruses [ICTV]: order: Mononegavirales; family: Paramixoviridae; subfamily: Orthoparamyxovirinae) [1–3]"
  2. Materials and Methods (Exclusion criteria): Could the authors clarify if there was a definition of a minimum number of participants in the studies of interest (i.e., n > 30)? REPLY: we clarified that, for seroprevalence studies, we implemented a preventive, arbitrary cut-off of 25 cases ("Studies providing seroprevalence data from less than 25 HCWs were included into the systematic review but arbitrarily removed from the meta-analysis").
  3. Materials and Methods (2.5.2 Meta-Analysis): For greater clarity, please include the formula used to calculate the confidence interval of seroprevalence with the software used. REPLY: Thank you for your suggestion: we amended (we should say rewrite) chapter 2.5.2 as follows: "

    Pooled prevalence rates for NiV-targeting antibodies were meta-analyzed through a random effect model (REM) that implemented the inverse variance method and a maximum likelihood estimator for calculation of tau2 (REML). Freeman Tuckey double arcsine transformation was implemented for the transformation of proportion before meta-analysis and preferred over logit transformation as considered more effective in dealing with samples of reduced and heterogenous size [46]. Finally, 95%CI of pooled estimates were calculated by means of the highly conservative Clopper Pearson approach [47]. REM approach was implemented and preferred over the fixed effects model as it is usually considered more effective in dealing with meta-analyses, including a reduced number of sampled studies and high heterogeneity across the source data [48,49]. Similarly, the REML was preferred over other methods (e.g. DerSimonian and Laird) as it excludes the summary effect parameter in its estimation of tau2 (unlike to DL), being therefore considered more effective in reducing residual bias [50]". Please take into account that, in order to cope with comments with other reviewers, we did redo the whole of calculations implementing different approaches for calculation of pooled estimates, with significant differences from the first round of revision (but no changes into the key messages to be delivered).

  4. Results (Figure 2): My suggestion is to make the current figure smaller and insert another one next to it (larger), highlighting the sampled geographic regions (i.e., state/province, city...). REPLY: Figure 2 was remade by taking into account the administrative divisions.
  5. Results: Please considerer writing “Forest plot” instead of “Forrest plot”. Reply: Thank you, we fixed it.
  6. Discussion: Could the authors discuss the serological diagnostic tests used in the selected studies: whether they are in-house or commercial tests; and if there is literature data on the laboratory accuracy of the tests (sensitivity, specificity...)? REPLY: the topic was specifically addressed by implementing a subgroup analysis by characteristics of antigens for EIA/ELISA test, and the main text was reworked accordingly: 

    "Focusing on seroprevalence studies, 2 of them were based on enzyme immunoassay [31,60], while 5 studies included data based on ELISA [23,61–63,67]: interestingly, both EIA studies, and three out of 5 ELISA-based relied on United States CDC-provided antigens [23,31,61,62], while the more recent studies from Ramachandran et al. [63], and Yadav et al. [67], deliberately implemented indigenous-derived antigens, with high sensitivity and specificity compared to CDC-provided ones (i.e. 100% and 83.3%, respectively) [69]".

  7. Discussion (Limitation): It is important to note that the high level of heterogeneity observed is a limiting factor. This variability may be associated with differences in population groups, detection methods, sample types, and other contributing factors. REPLY: thank you, we implemented this point in several iterations of the discussion section: "

    Third, as the studies span between 1998 and 2019 decades, diagnostic strategies have evolved over time, providing increasingly accurate and effective diagnostic options even in the limited-resource areas that have been particularly affected by the outbreaks included in the present systematic review with meta-analysis [4,72]. The very same reference antigens did evolve over time, as original specimens provided by United States CDC and derived from the original outbreaks from Malaysia and Singapore (i.e. NiV-A) have been replaced by indigenous derived antigens from NiV-B, with allegedly improved diagnostic performances [63,67,69], but also potential constraints when pooling and comparing corresponding seroprevalence data." ... "

    Fourth, we must stress the potentially limited representativity of the pooled sample. Seroprevalence studies gathered data on a total of 910 HCWs, while the total number of exposed professionals reported by studies on NiV outcomes was not actually provided by all parent studies. While the studies from Mounts et al. [60], Chan et al. [61], Gurley et al. [62], and Kumar et al. [23], reasonably provided a comprehensive report on the HCWs exposed to NiV during the documented outbreaks, the studies from Hsu et al. [31], Ramachandran et al. [63], and Sazzad et al. [68] included a more restricted sample, whose definition was more unclear. For example, the latter study includes a total of 328 cases, representing around 44.32% of total NiV cases included into the pooled analyses on the outcome of viral infection. As the status of HCWs was undoubtedly associated with a single case, a physician, who developed a fatal infection, while whether other HCWs were actually included into the sample remains undisclosed, a single HCW from this study was added to the pooled sample. Consequently, we cannot rule out that the very CFR we documented may be due to the oversampling of severe cases. In this regard, also in sero-prevalence studies on NiV a significant proportion of cases do not provide the total number of involved and/or occupationally exposed HCWs, rather focusing on subjects having been exposed to bodily fluids and respiratory specimens of affected cases, a definition that therefore does not specifically refers to HCWs, but that can include all individuals assisting NiV cases during their hospitalization [114–118]." "

    Fifth, studies were quite heterogeneous in terms of their design, and the meta-analysis included two case-control study, four cohort studies, and 10 cross-sectional studies. Cohort studies and cross-sectional studies are usually considered comparable in design, with the notable difference that in the cross-sectional design exposures and outcomes are measured at the same time (i.e. cross-sectionally), whereas in a cohort study outcomes are typically measured after the exposure/s has been measured (i.e. longitudinally) [45]. Due to the characteristics of immune responses to NiV, and to the dubious timing of individual exposure to NiV, the different underlying design has possibly led to some degree of eventual inaccuracy in pooled estimates."

Reviewer 2 Report

Comments and Suggestions for Authors

Based on the unclear study significance and significant methodological flaws outlined below, this manuscript should not be recommended for publication:

#Introduction
1. The authors have not clearly established the significance of the study in the Introduction section. Additionally, the length of the Introduction could be reduced, as it includes unnecessary details that do not directly contribute to the study’s focus.

#Methods

2. Search Strategy: The search strategy outlined in the manuscript has several serious limitations that likely contributed to the absence of studies from 2020-2024:

  • Narrow Search Terms: The search terms used were limited and may not have captured the full scope of relevant literature. Key health professionals (e.g., physicians, nurses, pharmacists, orthopedics specialists, etc.) were not included, and proximity operators were not used to broaden the search.
  • Geographic Limitations: The search did not incorporate specific country names, particularly from high-risk regions like South Asia and the Asia Pacific, where the Nipah virus is most prevalent. This likely explains the lack of relevant studies from these areas, especially after 2019.
  • Exclusion of Grey Literature: The search excluded grey literature, such as preprints, conference proceedings, and reports from public health organizations, which are valuable sources of recent research and may contain studies not yet published in peer-reviewed journals.

These issues likely resulted in missed studies and a potential gap in the review.

Recommendations:

  • Revise the search strategy to include a broader range of health professionals and use proximity operators to ensure comprehensive coverage.
  • Expand the geographic scope by incorporating specific country names, particularly from South Asia and the Asia Pacific regions.
  • Include grey literature sources, such as preprints and conference abstracts, to capture any relevant studies that may have been missed in peer-reviewed journals.
  • Include additional databases such as Web of Science, nursing databases (e.g., CINAHL)

3. Quality assessment tool selection: The authors have made a significant error in selecting the quality assessment tool for this review. The quality of observational studies should be evaluated using established and appropriate tools such as the Newcastle-Ottawa Scale (NOS) or ROBINS. However, the authors have inappropriately applied the OHAT tool, which is not designed for use with observational studies and is entirely unsuitable for this type of analysis.

4. Clarification on study designs: The manuscript does not clearly specify the different types of observational studies included (e.g., cohort, case-control, cross-sectional, etc.) in the inclusion criteria, study characteristics, or tables. Since each study design has its own strengths and limitations, combining them in a pooled analysis without accounting for these differences could lead to inaccurate conclusions. This lack of consideration raises concerns about the validity of the meta-analysis.

5. Clarification on data transformation: The authors report pooled prevalence rates for NiV-targeting antibodies were meta-analyzed through a random effect model, being then reported as point estimates with their 95%CIs.

However, it is important to clarify whether the authors applied a log, logit or arcsine transformation to the proportions prior to the meta-analysis. Transformations are typically necessary when analyzing proportions (such as prevalence rates) to stabilize variances, especially when the data includes extreme values (close to 0 or 1), ensuring the results are more reliable and appropriate for meta-analysis.

6. Heterogeneity classification: The authors cite two outdated references to define the I² thresholds for heterogeneity, claiming these as "current recommendations." However, these references are not recent, and the classification of heterogeneity has evolved in more recent literature. According to current Cochrane guidelines, the following thresholds are generally accepted: low (0-25%), moderate (25-50%), substantial (50-75%), and considerable (>75%).

§  https://handbook-5-1.cochrane.org/chapter_9/9_5_2_identifying_and_measuring_heterogeneity.htm

§  https://training.cochrane.org/msu-web-clinic-april-2023

7. Meta-analysis methodology: The authors do not specify which method was used for the meta-analysis, such as restricted maximum likelihood, the Knapp–Hartung standard-error adjustment, or the Hunter–Schmidt method. It is important to clarify this, as different methods can impact the results and their interpretation.

Additionally, in a meta-analysis of proportions, confidence intervals (CIs) represent the expected average estimate of all possible studies. However, the authors do not mention which method was used to calculate the CIs for the proportions. Common methods include the Wald, Wilson-Score, or Clopper-Pearson intervals, and the choice of method can influence the precision and validity of the reported CIs (Barker et al., 2021).

Author Response

Estimated Reviewer,

we did acknowledge your negative comments and we tried our best for improving the overall quality of the manuscript. Please see the following points: we would stress that we thank you for the comments you did share, as we're confident about the positive impact of your suggestions on the overall quality of the present paper.

Based on the unclear study significance and significant methodological flaws outlined below, this manuscript should not be recommended for publication:

COMMENT 1. The authors have not clearly established the significance of the study in the Introduction section. Additionally, the length of the Introduction could be reduced, as it includes unnecessary details that do not directly contribute to the study’s focus.

REPLY: the introduction was reduced in its total length, and we revised its final sections as follows: "In other words, even though the potential for the global spread of NiV still remains quite limited [5,23,25,26,28], it could represent a challenging effort for HCWs and healthcare systems [28,33,34], as recently stressed [35] not only from the health safety point of view, but also when considering the potential ethical issues. Due to the perceived but still latent risk that NiV may evolve from an emerging tropical disease to a potentially epidemic or even pandemic global threat [36–38], assessing how this pathogen did affect HCWs and healthcare systems during local outbreak may provide some guidance for all healthcare professionals and stakeholders. Therefore, this systematic review with meta-analysis aimed to gather and synthesize the existing evidence on the seroprevalence of NiV in HCWs involved in the management of index cases. Our research could provide a comprehensive understanding of the risks possibly associated with inpatient care during an outbreak, leading to developing strategies for improving preparedness, safety, and general support during NiV epidemics."

COMMENT 2: Search Strategy:The search strategy outlined in the manuscript has several serious limitations that likely contributed to the absence of studies from 2020-2024:

  • Narrow Search Terms: The search terms used were limited and may not have captured the full scope of relevant literature. Key health professionals (e.g., physicians, nurses, pharmacists, orthopedics specialists, etc.) were not included, and proximity operators were not used to broaden the search. Geographic Limitations: The search did not incorporate specific country names, particularly from high-risk regions like South Asia and the Asia Pacific, where the Nipah virus is most prevalent. This likely explains the lack of relevant studies from these areas, especially after 2019. Exclusion of Grey Literature: The search excluded grey literature, such as preprints, conference proceedings, and reports from public health organizations, which are valuable sources of recent research and may contain studies not yet published in peer-reviewed journals.

REPLY: in accord to your comments, we revise the search strategy to include a broader range of health professionals and when possible we implemented proximity operators to ensure comprehensive coverage. Moreover, we expand the geographic scope by incorporating specific country names, particularly from South Asia and the Asia Pacific regions, did include grey literature sources, such as preprints and conference abstracts (where available) and included two additional databases i.e. CINAHL and medrixiv. 

Following search strategies were implemented:

PubMed: "Nipah Virus"[Mesh] AND (("Indonesia" OR "Cambodia" OR "Timor" OR "Malaysia" OR "Philippines" OR "Singapore" OR "Thailand" OR "India" OR "Bangladesh") OR ("Health Personnel" [Mesh] OR "Allied Health Personnel" [Mesh] OR "healthcare worker*" OR "health care worker*" OR "nurs*" OR "work*" OR "occupational" OR "health professional*" OR "medical practitioner*" OR "medical doctor*" OR "nursing professional*" OR "midwifery professional*" OR "midwife" OR "paramedic*" OR "surgical technician*" OR "dentist*" OR "physiotherapist*" OR "laboratory technician*" OR "pathologist*" OR "medical assistant*" OR "ambulance officer*" OR "emergency medical technician*" OR "emergency paramedic*")).

Scopus: "Nipah" OR "Nipah virus" AND (("Indonesia" OR "Cambodia" OR "Timor" OR "Malaysia" OR "Philippines" OR "Singapore" OR "Thailand" OR "India" OR "Bangladesh") OR ("Health Personnel" OR "Allied Health Personnel" OR "healthcare worker*" OR "health care worker*" OR "nurs*" OR "work*" OR "occupational" OR "health professional*" OR "medical practitioner*" OR "medical doctor*" OR "nursing professional*" OR "midwifery professional*" OR "midwife" OR "paramedic*" OR "surgical technician*" OR "dentist*" OR "physiotherapist*" OR "laboratory technician*" OR "pathologist*" OR "medical assistant*" OR "ambulance officer*" OR "emergency medical technician*" OR "emergency paramedic*"))EMBASE: (‘nipah virus’/exp OR ‘nipah virus’ OR ‘nipah virus infection’) AND (‘health care personnel’ OR ‘occupational’ OR ‘work AND related’ OR ‘nursing staff’ OR ‘nurse’).

CINAHL: "Nipah” AND (("Indonesia" OR "Cambodia" OR "Timor" OR "Malaysia" OR "Philippines" OR "Singapore" OR "Thailand" OR "India" OR "Bangladesh") OR ("Health Personnel" OR "Allied Health Personnel" OR "healthcare worker*" OR "health care worker*" OR "nurs*" OR "work*" OR "occupational" OR "health professional*" OR "medical practitioner*" OR "medical doctor*" OR "nursing professional*" OR "midwifery professional*" OR "midwife" OR "paramedic*" OR "surgical technician*" OR "dentist*" OR "physiotherapist*" OR "laboratory technician*" OR "pathologist*" OR "medical assistant*" OR "ambulance officer*" OR "emergency medical technician*" OR "emergency paramedic*"))

EMBASE: (‘nipah virus’/exp OR ‘nipah virus’ OR ‘nipah virus infection’) AND (‘health care personnel’ OR ‘occupational’ OR ‘work AND related’ OR ‘nursing staff’ OR ‘nurse’ OR ‘occupational accident’ OR ‘paramedical personnel’ OR ‘health practitioner’ OR ‘midwife’ OR ‘physiotherapist’ OR ‘rescue personnel’ OR ‘laboratory personnel’ OR ‘medical assistant’ OR ‘dentist’ OR ‘pathologist’)

MEDRXIV: "Nipah" OR "nipah virus" AND ("health personnel" OR "allied health personnel" OR "healthcare worker*" OR “nurs*”)

Unfortunately, despite this editing, only two further studies were retrieved, one of them for the time period 2020-2024. After the extensive review of the additional papers retrieved from database analysis, we noticed that  a large proportion of studies, including both "grey" literature ones and local studies from competent bodies usually do not focus on regular HCWs, but rather on subjects performing care for individuals affected by NiV. The reason has been partially addressed in the paper from Sazzad et al. (2013) that we included in the present study after the required revisions: in some of the affected areas, particularly Bangladesh, the care for affected individuals may be shared by regular and "irregular" healthcare workforce. In fact, we were able to implement the paper from Yadav et al. (2022) as it included the detailed characteristics of single cases (supplementary material). On the contrary, we were unable to implement an otherwise interesting paper from Sankar et al. (BMC Proceedings 2021, 15(Suppl 11):17, a conference report) as it included only results from RT-qPCR without serological appraisal and not including any NiV+ case, it was also inconsistent with the aim for CFR analysis. 

Comment 3: Quality assessment tool selection: The authors have made a significant error in selecting the quality assessment tool for this review. The quality of observational studies should be evaluated using established and appropriate tools such as the Newcastle-Ottawa Scale (NOS) or ROBINS. However, the authors have inappropriately applied the OHAT tool, which is not designed for use with observational studies and is entirely unsuitable for this type of analysis.

Reply: Even though we did not agree with some of your comments (in fact, OHAT can be used for observational studies, see https://ntp.niehs.nih.gov/sites/default/files/ntp/ohat/pubs/handbookmarch2019_508.pdf pag. 35 and followings), we otherwise acknowledged that NOS (despite its original lack of specifically designed tools for cross-sectional studies) benefits from more extensive use and therefore the paper could in turn benefit from its implementation, so we did even though it changed quite substantially the eventual appraisal of the included studies. Please see the section:

"A summary of the quality assessment is provided in Table 3. In summary, the quality of most of included studies was appreciated as high or even very high, as 14 out of 16 studies had a cumulative score equals to or higher than 7 out of 9. Only the studies from Ching et al. [16], and Gurley et al. [62], were affected by several shortcomings in various domains, most notably when dealing with the representativity of the sample and the ascertainment of the baseline condition. However, despite the good quality of the reports, remaining studies were affected by some potential issues. For instance, the study from Mounts et al. [60] did not provide a clear explanation for the recruitment of sampled HCWs from the original series of occupationally exposed workers. Another possibly biased study, at least from the aims of the present report, was that of Ramachandran et al. [63], as it was initially designed to provide a description of individuals possibly exposed to an index case, including the 26 sampled HCWs. In fact, some degree of bias in terms of selection bias similarly affected other reports on NiV outbreaks, and more precisely, the studies from Arunkumar et al. [32], Chakraborty et al. [64], and Chadha et al. [30], as their design does not guarantee that all potentially exposed HCWs were ultimately included in the study. Similarly, not only the study from Chadha et al. [30], but also the study from Nikolay et al. [65] did not provide accurate information on the outcome of sampled HCWs. Nonetheless, it is otherwise important to stress that the report from Nikolay et al. [65] similarly to the study from Ramachandran et al. [63] was not designed for an accurate description of NiV cases from healthcare settings, and these characteristics did potentially affect data reporting.

Comment 4. Clarification on study designs:The manuscript does not clearly specify the different types of observational studies included (e.g., cohort, case-control, cross-sectional, etc.) in the inclusion criteria, study characteristics, or tables. Since each study design has its own strengths and limitations, combining them in a pooled analysis without accounting for these differences could lead to inaccurate conclusions. This lack of consideration raises concerns about the validity of the meta-analysis.

Reply: we sincerely apologize for this very naïve mistake and we revised both material and methods, results and tables for stressing this very important issue (Results: "Regarding their underlying design, the majority of sampled studies had a CS design (10, 62.50%), while four studies had a CH design (25.0%), and two studies a CC design (12.50%)" Discussion: "Fifth, studies were quite heterogeneous in terms of their design, and the meta-analysis included two case-control study, four cohort studies, and 10 cross-sectional studies. Cohort studies and cross-sectional studies are usually considered comparable in design, with the notable difference that in the cross-sectional design exposures and outcomes are measured at the same time (i.e. cross-sectionally), whereas in a cohort study outcomes are typically measured after the exposure/s has been measured (i.e. longitudinally) [45]. Due to the characteristics of immune responses to NiV, and to the dubious timing of individual exposure to NiV, the different underlying design has possibly led to some degree of eventual inaccuracy in pooled estimates."

Comment 5: Clarification on data transformation: The authors report pooled prevalence rates for NiV-targeting antibodies were meta-analyzed through a random effect model, being then reported as point estimates with their 95%CIs. However, it is important to clarify whether the authors applied a log, logit or arcsine transformation to the proportions prior to the meta-analysis. Transformations are typically necessary when analyzing proportions (such as prevalence rates) to stabilize variances, especially when the data includes extreme values (close to 0 or 1), ensuring the results are more reliable and appropriate for meta-analysis.  Meta-analysis methodology:The authors do not specify which method was used for the meta-analysis, such as restricted maximum likelihood, the Knapp–Hartung standard-error adjustment, or the Hunter–Schmidt method. It is important to clarify this, as different methods can impact the results and their interpretation. Additionally, in a meta-analysis of proportions, confidence intervals (CIs) represent the expected average estimate of all possible studies. However, the authors do not mention which method was used to calculate the CIs for the proportions. Common methods include the Wald, Wilson-Score, or Clopper-Pearson intervals, and the choice of method can influence the precision and validity of the reported CIs (Barker et al., 2021).

Reply: As for point 4, we sincerely apologize as you did point a very important issue that we improperly handled. As we included two further studies into the pooled sample, we did redo all the calculation as summarized in the totally rewritten section 2.5.2 

Pooled prevalence rates for NiV-targeting antibodies were meta-analyzed through a random effect model (REM) that implemented the inverse variance method and a maximum likelihood estimator for calculation of tau2 (REML). Freeman Tuckey double arcsine transformation was implemented for the transformation of proportion before meta-analysis and preferred over logit transformation as considered more effective in dealing with samples of reduced and heterogenous size [46]. Finally, 95%CI of pooled estimates were calculated by means of the highly conservative Clopper Pearson approach [47]. REM approach was implemented and preferred over the fixed effects model as it is usually considered more effective in dealing with meta-analyses, including a reduced number of sampled studies and high heterogeneity across the source data [48,49]. Similarly, the REML was preferred over other methods (e.g. DerSimonian and Laird) as it excludes the summary effect parameter in its estimation of tau2 (unlike to DL), being therefore considered more effective in reducing residual bias [50].

Comment 6. Heterogeneity classification:The authors cite two outdated references to define the I² thresholds for heterogeneity, claiming these as "current recommendations." However, these references are not recent, and the classification of heterogeneity has evolved in more recent literature. According to current Cochrane guidelines, the following thresholds are generally accepted: low (0-25%), moderate (25-50%), substantial (50-75%), and considerable (>75%).

Reply: again, we sincerely thank you for this important comment. The thresholds were revised consistently with the recommended guidelines (i.e. Therefore, the 95%CIs of the point I2 estimates were also calculated and reported. In accordance with current recommendations, heterogeneity was considered low for I2 values ≤ 25%, moderate for I2 values ≥ 26% and < 50%, substantial for I2 values ranging between 50% and 75%, and considerable for estimates > 75% [44,52,53]).

Eventually we would thank you for the accurate, rigorous, but also propositive comments. Again, we would stress that we tried our best for improving the paper in accord with your suggestion but we are also open to making further adjustments where deemed appropriate.

Reviewer 3 Report

Comments and Suggestions for Authors

Review report for article number 3321292 in MDPI viruses.

The authors have written a systematic review on the risk of Nipah virus seroprevalence in health care workers with meta-analysis and claimed the reasonable conclusion of basic preventive measures, including the use of personal protective equipment (PPE) and appropriate isolation of incident cases with physical distancing, may be quite effective in avoiding the occurrence of new infections among Health care.

Authors have done an in-depth study on the Nipah virus seroprevalence cases and I am happy to accept the current review article in the Journal of Viruses in MDPI.

The authors have addressed the meta-analysis and systematic review on Nipah virus, a zoonotic pathogen capable of causing human outbreaks with a high case fatality rate and authors has critically evaluated the available evidence on NiV infections among healthcare workers (HCWs).

The Nipah virus poses a significant threat to HCWs due to its zoonotic nature and potential for human-to-human transmission. Despite the documented outbreaks, there remains a critical gap in the literature exploring the specific role of PPE and physical distancing in preventing NiV infections among HCWs. Given the high case fatality rates associated with the virus, implementing and rigorously adhering to these measures could drastically reduce transmission risks. Addressing this gap through targeted studies would not only validate the effectiveness of these interventions but also provide evidence-based guidelines to enhance the safety of HCWs during this virus outbreaks, ensuring better preparedness and response strategies.

Nipah virus is highly fatal, and limited knowledge and resources in low- and middle-income countries often lead to unintentional neglect of basic preventive measures, such as the proper use of PPE and avoiding the sharing of basic hospital equipment. This study provides a valuable contribution to the existing literature by emphasizing the importance of these fundamental preventive measures, addressing a critical gap in the understanding of practical infection control strategies for managing Nipah virus outbreaks.

Based on my understanding and expertise in the field, the authors have gathered a substantial amount of information on the topic, which appears sufficient for the current study.

As per my knowledge the authors made the reasonable conclusions from the study that taking basic preventive interventions like using the PPE and maintaining the physical distancing from others will be better in avoiding the virus among the healthcare workers.

I suggest the authors clearly outline the significance of the current study in a dedicated paragraph to better highlight its contributions and relevance to the field.

Author Response

Estimated Reviewer,

thank you for your very positive comments and appraisal.

We've revised the paper accordingly, and implemented a series of subsections into the discussion, including the following one, "implications for daily practice" whose content was revised as follows:

During the first documented NiV outbreaks, including the first occupational health studies, in most cases, episodes of interhuman transmission of this pathogen were not documented, and nearly all infections were associated with the exposure of human cases to affected animals or fruits and vegetables contaminated with animal fluids [2,5–7,7–9,12–14]. On the contrary, outbreaks in the Indian subcontinent were characterized by documented inter-human transmission [18,21,32,35,66,88–90], stressing the pandemic potential that has been allegedly associated with NiV, particularly after the inception of the SARS-CoV-2 pandemic [36]. Even though NiV still appears confined to areas corresponding with the documented habitats of the main host (i.e., bats in the Pteropus genus), the inter-human transmission of NiV has resulted in significant implications for all professionals caring for affected patients. NiV has not only emerged as a potential occupational risk agent [28,62], but also as a nosocomial one, affecting HCWs as well as patients and individuals visiting and/or assisting friends and relatives during their institutionalization [60,62,91]. Following the blueprint learned during the SARS-CoV-2 and mpox pandemic [34,92,93], but also when dealing with more “conventional” viral pathogens such as varicella-zoster virus and measles [94–99], designing, tailoring, and properly implementing appropriate preventative occupational health policies represent therefore a very effective way for achieving at the same time HCWs and patient safety. Occupational physicians (OP), who are the medical professionals responsible for health prevention and promotion in the workplaces, actively contribute to the prevention of biological risk agents by applying and tailoring official recommendations [100,101], by either providing or promoting appropriate immunizations [101–105], by contributing to the identification and promotion of appropriate PPE [34], and specifically tailored post-exposure prophylaxis procedures [106]. In this regard, our study suggests that

Interestingly enough, implementing the face mask mandate in patients, caregivers, and HCWs, and avoiding unnecessary interactions between patients, their caregivers, and healthcare personnel have been identified as quite effective measures for countering the spreading of NiV in enclosed hospital settings [21].

Far from sharing a potentially deleterious false sense of security, our results suggest that by applying rigorous infection prevention and control practices, even in healthcare settings characterized by reduced availability of medical resources, all HCWs (i.e., all medical professionals, nurses and allied medical professionals) could safely perform their daily duties and care for NiV-infected patients. Interestingly, effective prevention of NiV was achived even before the COVID-19 pandemic did raise the level of attention towards infection control protocols, stressing that NiV can be faced in a very cost-effective way [21,90].

Nonetheless, interhuman spreading and subsequent epidemic and even pandemic potential of NiV should not be underestimated. For instance, data from previous outbreaks hint that NiV could quite effectively spread in hospital settings where index cases were not properly isolated, affecting all bystanders not benefiting from infection control protocols, mask-wearing policies and proper room ventilation [62,107,108]. In this regard, the study from Gurley et al. [62] provided further significant insight. While no one among the 105 sampled HCWs exhibited IgM-class antibodies, 2 nursing students who reported having changed bed sheets for index cases had IgG-class antibodies against NiV: even though neither student reported any unprotected exposure to bodily fluids of inpatients with documented NiV infection, their use of PPE (particularly masks and gloves) was not seemingly accurate.

Thank you again for your positive appraisal,

on the behalf of all Authors,

MR